

# Classification of *Curcuma longa* and *Curcuma zanthorrhiza* using transfer learning

Agus Pratondo[1], Elfahmi Elfahmi[2] and Astri Novianty[3]

[1] Department of Multimedia Engineering, School of Applied Sciences, Telkom University, Bandung, West Java, Indonesia
[2] Department of Pharmaceutical Biology, School of Pharmacy, Bandung Institute of Technology, Bandung, West Java, Indonesia
[3] Department of Computer Engineering, School of Electrical Engineering, Telkom University, Bandung, West Jawa, Indonesia

## ABSTRACT

*Curcuma longa* (turmeric) and *Curcuma zanthorrhiza* (temulawak) are members of the *Zingiberaceae* family that contain curcuminoids, essential oils, starch, protein, fat, cellulose, and minerals. The nutritional content proportion of turmeric is different from temulawak which implies differences in economic value. However, only a few people who understand herbal plants, can identify the difference between them. This study aims to build a model that can distinguish between the two species of *Zingiberaceae* based on the image captured from a mobile phone camera. A collection of images consisting of both types of rhizomes are used to build a model through a learning process using transfer learning, specifically pre-trained VGG-19 and Inception V3 with ImageNet weight. Experimental results show that the accuracy rates of the models to classify the rhizomes are 92.43% and 94.29%, consecutively. These achievements are quite promising to be used in various practical use.

## INTRODUCTION

Computer vision has played an important role in various fields, including biomedical engineering (*Rizqyawan et al., 2020*; *Pratondo et al., 2020*; *Zunair & Hamza, 2020*; *Rizqyawan et al., 2022*), smart farming, and agriculture (*Rajasekaran et al., 2020*; *Jahanbakhshi et al., 2021*; *Pratondo & Bramantoro, 2022*; *Pratondo & Harahap, 2022*). The use of smart technology in those fields have been widely used to support many activities, *e.g.*, seed classification, disease detection, automatic watering, and quality assessment (*Pratondo, Ong & Chui, 2014*; *Budiwati et al., 2021*; *Jong & Pratondo, 2011*; *Pratondo, 2010*). Recently, classification and detection using machine learning have broadly been utilized to carry on many tasks. Several diseases on root, plan, and fruit can be identified using a machine learning approach. Similarly, tasks on classifying trees, fruits, and vegetables are also possible, including turmeric.

Temulawak is a native Indonesian plant that looks similar to turmeric, *i.e.*, the plant has light yellow skin. As a monocot plant, it does not have a taproot. The root that is owned is

Corresponding author
Agus Pratondo,
pratondo@gmail.com

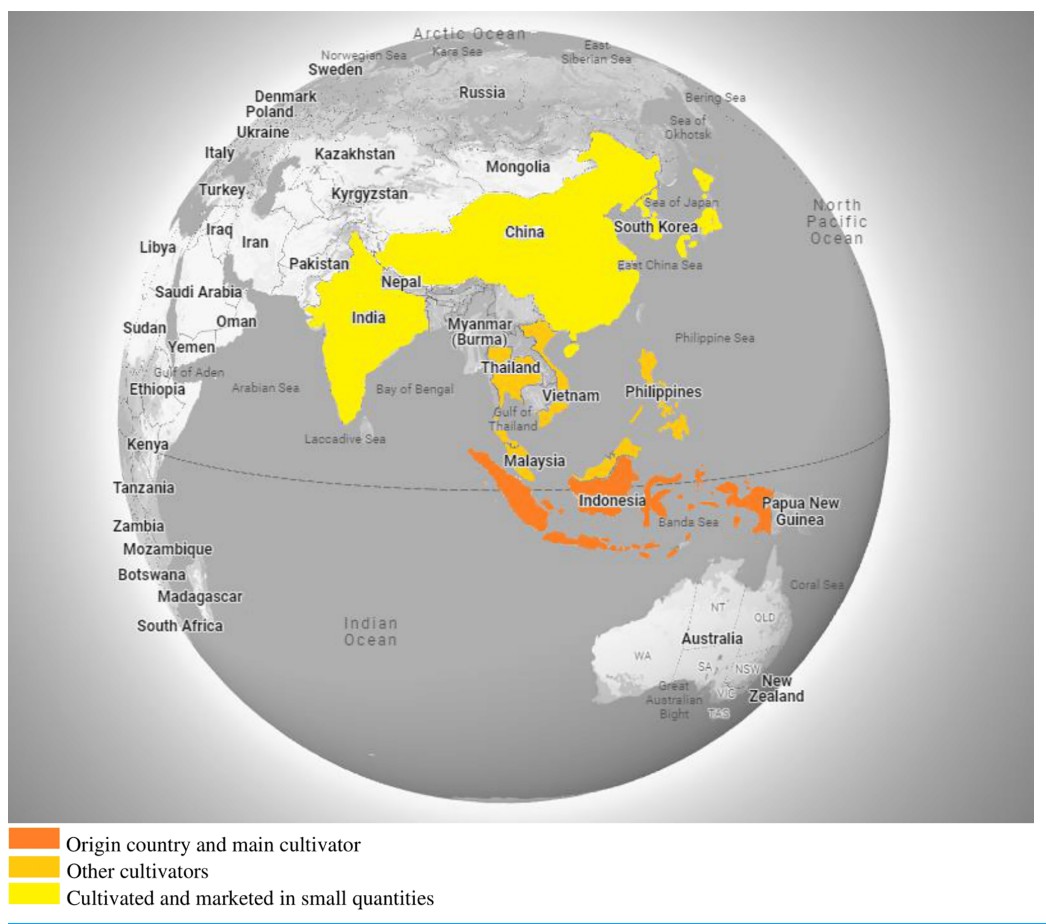

Origin country and main cultivator
Other cultivators
Cultivated and marketed in small quantities

**Figure 1** **Geographic distribution of *Curcuma zanthorrhiza*.**

the rhizome root. The rhizome is the part of the stem that is located underground. A rhizome is also referred to as a root tuber or stem tuber. Temulawak grows a lot in forests, especially teak forests, along with other rhizome plants. But now, temulawak is widely cultivated in the highlands due to its economic value compared to the original turmeric. The geographic distribution of the *Curcuma Zanthorrhiza* is shown in Fig. 1.

Temulawak is a popular medicinal plant in Indonesia that has been used traditionally since a long time ago for health maintenance and treatment of many diseases. It is used as a component of jamu (Indonesian herbal medicines system) (*Elfahmi, Woerdenbag & Kayser, 2014*). The traditional uses of this plant have also been confirmed and proven by scientific researchers for its pharmacological activities such as antibacterial, anti-inflammatory, antioxidative, nephroprotective, antitumor, neuroprotective, hepatoprotective, and other activities (*Elfahmi, Woerdenbag & Kayser, 2014*). The chemical contents of temulawak have been also investigated. It has been reported to contain terpenoid compounds such as α–turmerone, β–turmerone α–curcumene, arturmerone, xanthorrhizol, germacrene, β–curcumene, β–sesquiphellandrene, curzerenone; curcuminoids such as curcumin, monodemethoxy curcumin, bis-demethoxy curcumin, octahydrocurcumin, 1-hydroxy-1,7-bis(4-hydroxy-3-methoxypheny1)-6-

heptene3,5-dione, dihydrocurcumin, and hexahydrocurcumin (*Rahmat, Lee & Kang, 2021*).

As a species from the same genus as *Curcuma xanthorrhiza*, turmeric (*Curcuma longa*) also contains curcuminoid as a dominant compound, as well as other compounds. If it is compared to turmeric, temulawak is more often found as a component of jamu for health purposes. Xanthorrhizol is a typical compound that exists in temulawak that is rarely found in *Curcuma longa* and other species. This compound has been also reported to be responsible for the pharmacological activity of the *Curcuma xanthorrhiza* plant. Therefore temulawak has a very potent plant for herbal medicines. In the application of temulawak for herbal medicines and other purposes, it is very important to identify the plant that could be differentiated from other species (*Elfahmi, Woerdenbag & Kayser, 2014*). Several techniques in the field of biology and pharmacy have been developed to determine and identify this plant. The intervention of artificial intelligence, such as deep learning techniques, is innovative for this purpose. Applying computational technology to various fields leads to a better life (*Mutiara, Hapsari & Pratondo, 2019*; *Pratondo et al., 2020*; *Rizqyawan et al., 2020*).

Several studies on turmeric classification using an artificial intelligence approach have been initiated. *Rajasekaran et al. (2020)*, *Kuricheti & Supriya (2019)* employed this approach to detect the turmeric plant diseases. Generally, plant disease can be detected using various machine learning algorithms. The use of deep learning has been investigated by *Jahanbakhshi et al. (2021)*. They found that the position, chrominance, structure, and size of plant leaves can be handled using deep learning (*Jahanbakhshi et al., 2021*). *Khrisne & Suyadnya (2018)* utilized the VGGNet-like network to recognize several herbs and spices. *Andayani et al. (2020)* used stomata microscopic images and applied deep neural networks to classify curcuma herbal plants.

Various studies related to classification and detection have been present in the literature, including the detection and classification of turmeric. However, the classification of the turmeric variant itself has not yet been investigated, particularly regular turmeric (*Curcuma longa*) and temulawak (*Curcuma zanthorrhiza*). Although both are variants of turmeric, each contains different chemicals to support the pharmaceutical industry. As a result, each has a different economic value in the market. Unfortunately, people are often confused about the difference between the two due to their similarity, as shown in Fig. 2. This study aims to help people with difficulty recognizing these two turmeric variants using a deep neural network with pre-trained weights. Images that are suspected to be turmeric will be further classified into two classes, namely, regular turmeric (*Curcuma longa*) and temulawak (*Curcuma zanthorrhiza*).

The discussions of *Curcuma longa* (turmeric) and *Curcuma zanthorrhiza* classification will be presented in the rest of this article in the following systematic manner. "Materials and methods" will discuss the methods used in the classification of *Curcuma longa* (turmeric) and *Curcuma zanthorrhiza* by taking the representation of two algorithms from the classical algorithms and two modern algorithms. Furthermore, the preparation of data sets, experimental settings, and experimental results are "Discussions" will elaborate on the results and discuss possible improvements that can be made for further research. Finally,

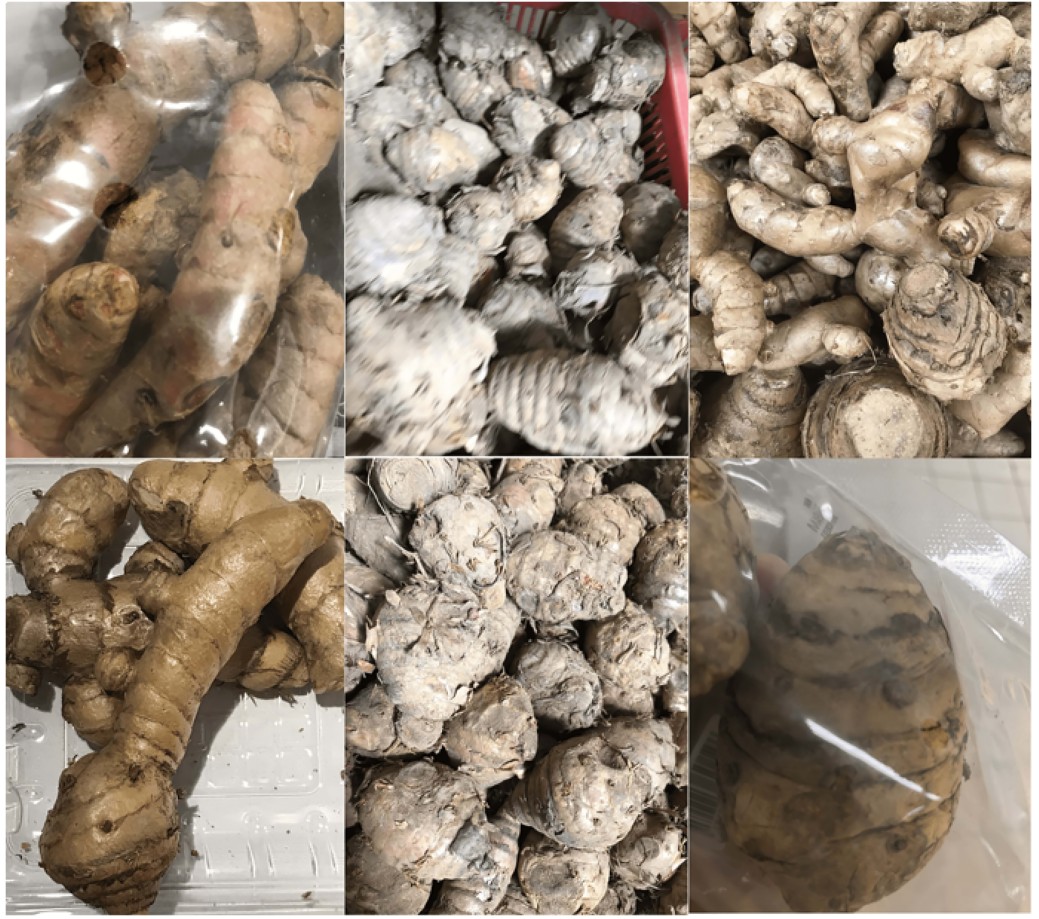

**Figure 2 Sample images of *Curcuma longa* (top) and *Curcuma zanthorrhiza* (bottom).**

conclusions about the experimental results and discussion are presented briefly in
"Conclusions".

# MATERIALS AND METHODS

In this research, curcuma image classification is conducted by building a model generated
from machine learning algorithms. Recent studies have shown that the use of deep learning
can handle agricultural image classification quite well (*Zheng et al., 2019*). For
benchmarking, experiments using traditional machine learning are also carried out, and
the results are employed as the baseline.

## Classic machine learning

The use of traditional classification algorithms is intended to obtain the baseline for
curcuma classification. Several well-known algorithms are available. Two algorithms are
employed, namely the $k-$nearest neighbours ($k$-NN) and the support vector machines
(SVM). The first one is chosen since it is commonly used due to its simplicity and accuracy.
At the same time, the SVM existed later and also be recognized well for its robustness

before the deep learning era (*Bishop, 1995*; *Breiman et al., 2017*; *Duda, Hart & Stork, 2012*; *Ham & Kostanic, 2000*; *Haykin, 2010*).

It is important to note that these classical machine learning algorithms are used for the baseline. The vector features are built by converting an RGB to a grayscale image with a specific size. All pixels in the grayscale image are used as features.

### k–nearest neighbours (k–NN)

The $k-$NN was first developed by Evelyn Fix and Joseph Hodges in 1951 (*Fix & Hodges, 1951*). The $k-$NN algorithm works by taking a number of $k$ closest data (neighbours) as a reference to determine the class of new data. This algorithm classifies data based on similarity to nearest data which can be summarized as follow (*Bishop, 1995*; *Duda, Hart & Stork, 2012*).

1. Determine the number of neighbors ($k$) that will be used for class determination considerations.
2. Calculate the distance from the new data to each data point in the dataset.
3. Take a number of $k$ data with the closest distance, then determine the class of the new data based on majority voting.

### Support Vector Machines (SVM)

Support Vector Machine (SVM) works by finding the best hyperplane or separator function to separate classes (*Cristianini & Shawe-Taylor, 2000*; *Géron, 2019*). The SVM algorithm has a well-established mathematical concept and basis, so it has become a popular algorithm. This algorithm can be used for classification and regression. In contrast to KNN, which needs to set one parameter only ($k$), the SVM has several parameters that need to be considered as follows.

- C: a positive number that indicates the strength of the regularization.
- Kernel: a family of functions that are used to transform the data from the lower dimension to the higher one.
- Gamma: a specific parameter for particular kernels (namely: radial basis function, polynomial and sigmoid).
- Degree: indicates the degree of the polynomial kernel function.

## Deep learning

Deep learning originally came from artificial neural networks, which are constantly improving. The earlier model of artificial neural networks (ANN) was a single-layer perceptron and later developed into multilayer perceptrons. The ANN became more popular after deep learning was introduced. In deep learning, the number of layers in neural network structure increases significantly compared to the traditional artificial neural networks (*Gulli & Pal, 2017*; *Abadi et al., 2016*; *Goodfellow, Bengio & Courville, 2016*; *Simonyan & Zisserman, 2014*; *He et al., 2016*).

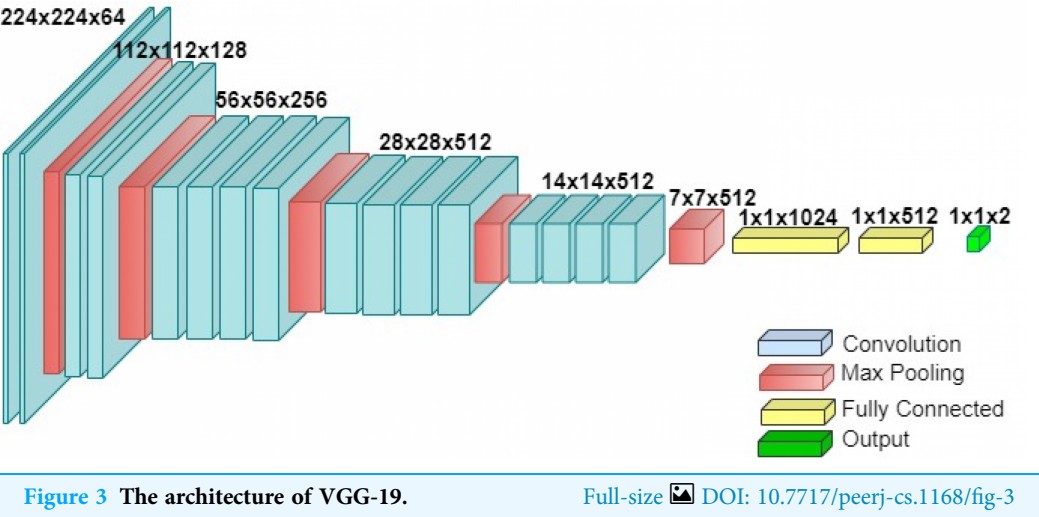

**Figure 3  The architecture of VGG-19.**

Since the number of layers is high, training time for the network is a tedious task. Fortunately, recent studies show that the weight for the earlier layers can be adopted from a previously well-trained model, which is commonly known as transfer learning. Several models with pre-trained weights are already investigated and available in the literature. Two pre-trained models are selected in this study, namely, the pre-trained VGG-19 and Inception V3 with ImageNet weight. The first model is chosen due to its simplicity which consists of only 19 layers, while the second is more modern and consists of more complex layers.

### VGG-19

VGG-19 is a convolutional neural network proposed by Visual Geometry Group, Department of Engineering Science, University of Oxford, Oxford, UK. The architecture of VGG-19 consists of 19 layers, *i.e.*, 16 convolutional layers and three fully connected layers. The architecture of the VGG-19 is shown in Fig. 3. This model is chosen because the architecture is simpler compared to the recent architecture in deep learning. The input size for this model is 224 × 224.

### Inception V3

The Inception V3 is the third edition of the Inception of Google's Inception Convolutional Neural Network. It was introduced in the ImageNet Recognition Challenge (*Szegedy et al., 2015*; *Tan & Le, 2019*). The architecture of the Inception V3 is shown in Fig. 4. As can be seen, it has more layers compared to the VGG-19. (*Shlens, 2016*).

### Experiment settings

The experiments are conducted using a dataset of *Curcuma longa* and *Curcuma zanthorrhiza*. The dataset is obtained from the various markets using a mobile phone. Curcuma images are taken randomly in various locations, both modern and traditional markets. In addition, the curcumas are captured in the same situation as they are presented in the market so that some are open while the others are wrapped in plastic, as shown in Fig. 2. Furthermore, we leave the level of wetness of the curcuma as it is, *i.e.*, some are still

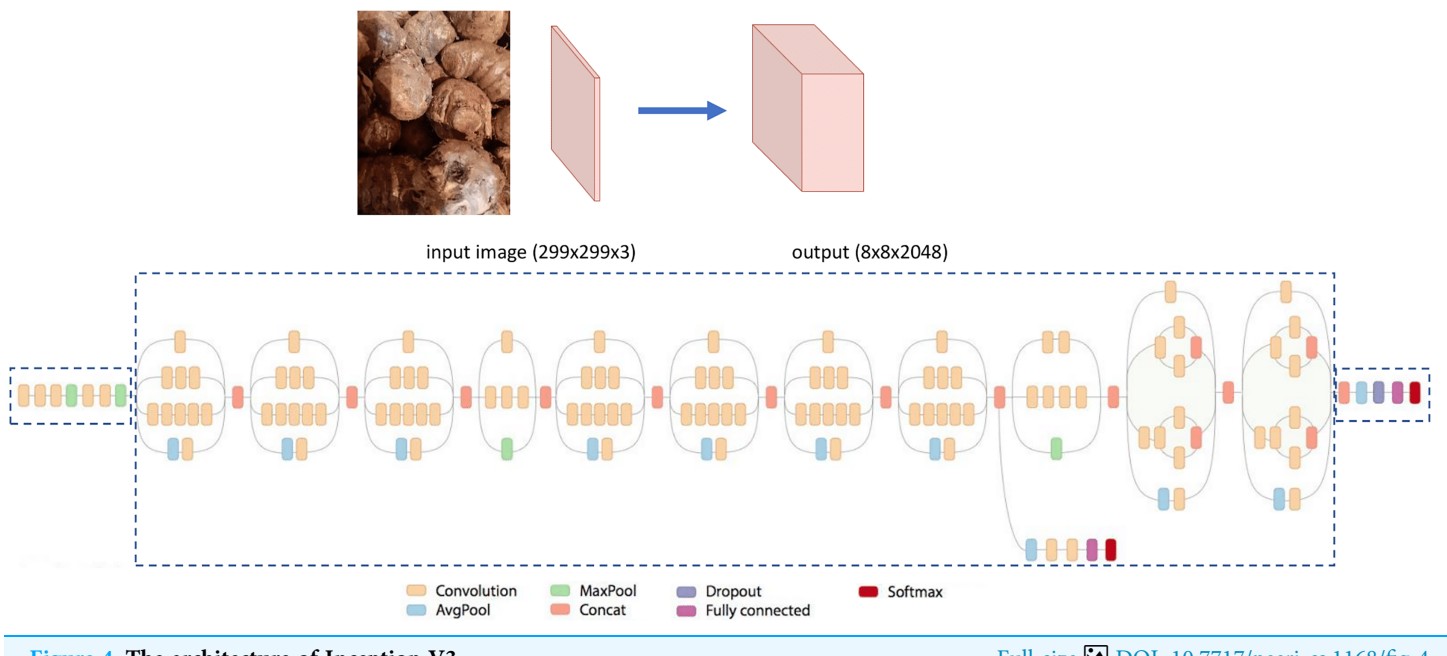

**Figure 4 The architecture of Inception V3.**

wet while the others are dry. The amount of obtained images of *Curcuma longa* and *Curcuma zanthorrhiza* are 396 and 251, respectively.

Subsequently, the obtained image is fed to the classification algorithm by dividing it into ten groups for cross-validation. Iteratively, one group is used as testing data, and the rest of the groups are used for training data. Each model of the classification algorithm will be tested with ten different groups, and the final accuracy is obtained by taking the average accuracy of the entire test. Because the curcuma image dataset is imbalanced for both classes, the dataset is divided into ten groups using stratified $k$ cross-validation. It ensures that the number of class proportions in each group is close to or equal to the class proportion in the entire dataset.

Due to the limited number of images used for the training in deep learning, we employ image augmentation, which is an efficient technique for training in deep learning models on limited data sets. A number of image augmentation parameters are set to simulate image variations in the training set, *i.e.*, rotation, zoom, shear, flip, and the mode to fill the blank pixel generated during augmentation. The augmentation is performed using ImageDataGenerator from Keras, and the value of the parameters are *rotation_range* = 15, *zoom_range* = 0.2, *shear_range* = 0.2, *horizontal_flip* = *True*, *vertical_flip* = *True*, and *fill mode* = 'nearest'. The number of augmented images used during the experiment was 8,000, or about 12.4 times the original images in the dataset. A number of images generated after augmentation are shown in Fig. 5.

## RESULTS

We performed curcuma classification on several algorithms. Firstly, we employed the $k-$NN by iterating the value of $k$ from 1 to 9. The even number was skipped to avoid

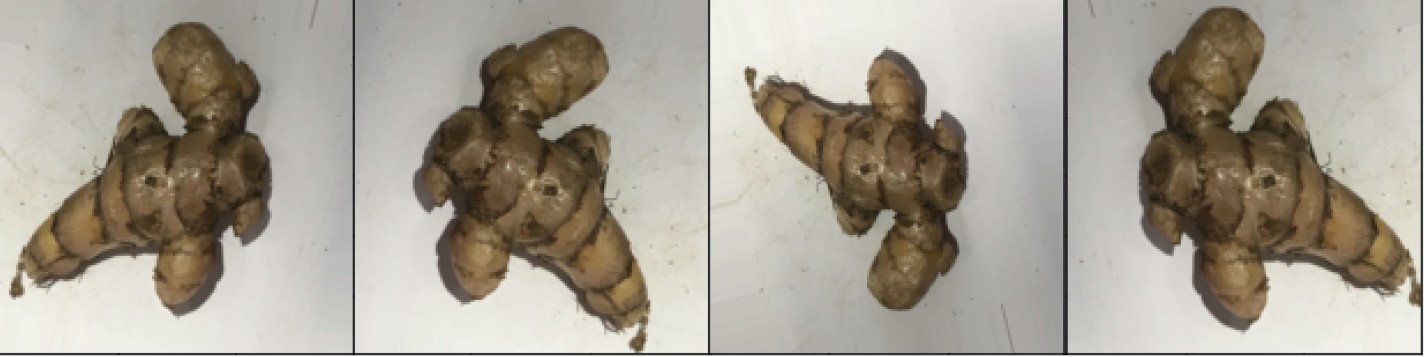

**Figure 5 Image augmentation on a *Curcuma zanthorrhiza*.**

| Testing fold | Training fold | Accuracy | | | | | Average |
|---|---|---|---|---|---|---|---|
| | | k = 1 | k = 3 | k = 5 | k = 7 | k = 9 | |
| 1 | 2–9 | 81.54 | 75.39 | 80.00 | 76.92 | 75.39 | 77.85 |
| 2 | 1, 3–10 | 76.92 | 80 | 81.54 | 83.08 | 80.00 | 80.31 |
| 3 | 1-2, 4–10 | 86.15 | 78.46 | 80.00 | 78.46 | 76.92 | 80.00 |
| 4 | 1-3, 5–10 | 80.00 | 83.08 | 84.62 | 81.54 | 81.54 | 82.16 |
| 5 | 1-4, 6–10 | 72.31 | 72.31 | 73.85 | 73.85 | 72.31 | 72.92 |
| 6 | 1-5, 7–10 | 81.54 | 76.92 | 81.54 | 80.00 | 75.39 | 79.08 |
| 7 | 1-6, 8–10 | 73.85 | 73.85 | 72.31 | 75.39 | 75.39 | 74.16 |
| 8 | 1-7, 9–10 | 79.69 | 75.00 | 75.00 | 81.25 | 78.13 | 77.81 |
| 9 | 1-8, 10 | 76.56 | 73.44 | 78.13 | 75.00 | 75.00 | 75.63 |
| 10 | 1-9 | 87.5 | 84.38 | 84.38 | 82.81 | 78.13 | 83.44 |
| Average | | 79.61 | 77.28 | 79.14 | 78.83 | 76.82 | 78.34 |

**Table 1 Experimental results using the k-NN.**

drawing during majority voting in binary classification. Then, the $k$ with the highest performance was taken and utilized for comparison to other algorithms. The performance using the $k-$NN is listed in Table 1.

It can be seen that $k = 1$ generates the best accuracy by achieving 76.91% accuracy, and the results are utilized for comparison in Table 2, where all results of the selected machine learning algorithms are presented.

We also perform curcuma classification using the SVM, one of the powerful traditional classification algorithms. Furthermore, two deep neural network architectures, namely VGG-19 and Inception V3, are also employed to classify the curcuma images. The deep neural networks are executed two times, *i.e.*, with and without pre-trained weights. Several parameters during experiments are set identically among the deep neural networks, *i.e.*, *epochs* = 40, *steps_per_epoch* = 20, and *batch_size* = 10. Experimental results using SVM and deep neural networks are shown in Table 2.

**Table 2 Experimental results on a number of the classification algorithm.** The best results from the k-NN in one are presented here.

| Fold for | | Accuracy | | | | | |
|---|---|---|---|---|---|---|---|
| Testing | Training | k-NN | SVM | VGG-19 (RI) | VGG-19 (PW) | Inc.V3 (RI) | Inc.V3 (PW) |
| 1 | 2–9 | 81.54 | 78.46 | 61.54 | 93.85 | 61.54 | 95.39 |
| 2 | 1, 3–10 | 76.92 | 75.39 | 61.54 | 92.31 | 61.54 | 100.00 |
| 3 | 1-2, 4–10 | 86.15 | 66.15 | 61.54 | 93.85 | 61.54 | 92.31 |
| 4 | 1-3, 5–10 | 80.00 | 64.62 | 61.54 | 93.85 | 47.69 | 95.38 |
| 5 | 1-4, 6–10 | 72.31 | 66.15 | 61.54 | 92.31 | 61.54 | 92.31 |
| 6 | 1-5, 7–10 | 81.54 | 61.54 | 61.54 | 89.23 | 67.69 | 89.23 |
| 7 | 1-6, 8–10 | 73.85 | 70.78 | 60.00 | 90.77 | 60.00 | 92.31 |
| 8 | 1-7, 9–10 | 79.69 | 75 | 60.94 | 85.94 | 60.94 | 93.75 |
| 9 | 1-8, 10 | 76.56 | 78.13 | 60.94 | 95.31 | 73.44 | 98.44 |
| 10 | 1-9 | 87.5 | 67.19 | 60.94 | 96.88 | 60.94 | 93.75 |
| Average | | 79.61 | 70.34 | 61.20 | 92.43 | 61.29 | 94.29 |

**Note:**
RI, Random initialization; PW, Pre-trained weights.

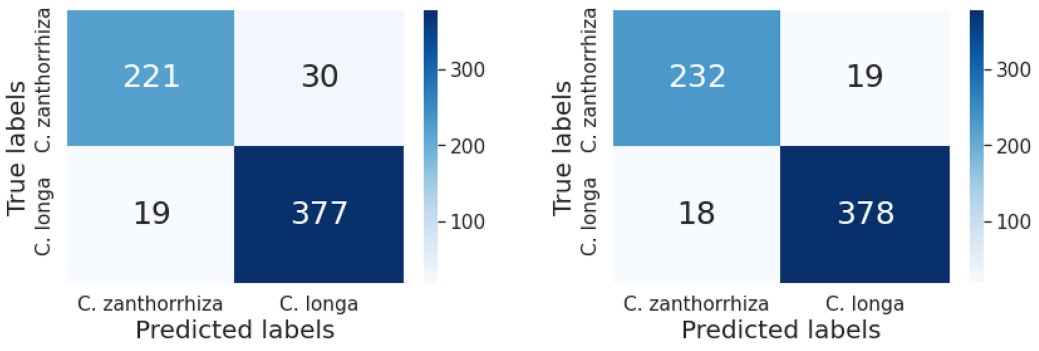

**Figure 6 Confusion matrix of models generated from (left) VGG-19 and (right) Inception V3.**

To elaborate on the performance of the classification, the confusion matrix for the VGG-19 and the Inception V3 using pre-trained weights are also recorded as shown in Fig. 6.

## DISCUSSIONS

Image classification on *Curcuma longa* and *Curcuma zanthorrhiza* using traditional machine learning algorithms produce models with low accuracy. Using the $k-$NN, the accuracy ranged from 76.82% to 79.61% for $k = 1, 3, 5,$ and 9, as shown in Table 1. As can be seen, the best results were obtained when $k = 1$. Similarly, the model using SVM provided lower accuracy than the model using $k-$NN, *i.e.*, 70.34%.

It is possible to employ well-hand-designed features commonly used before the deep learning era; however, building these features is not an easy task. In contrast, most current projects use neural network-based feature extraction with convolution. It is also possible to use the initial layers of the deep learning architecture as feature extractors. The discussion

**Table 3 Recall and precision using VGG-19 and Inception V3 with pre-trained weights.**

| Model | TP | TF | FP | FN | Precision | Recall |
|---|---|---|---|---|---|---|
| VGG-19 | 221 | 377 | 30 | 19 | 88.0% | 92.1% |
| Inception V3 | 232 | 378 | 19 | 18 | 92.4% | 92.8% |

will probably continue broadly regarding the feature extraction options, the number of layers that should be selected, *etc.* Therefore, we prefer using the simplest feature, *i.e.*, the raw pixels of the grayscale image, for the *k*-NN and SVM classifiers.

The experimental results from the traditional machine learning algorithms are subsequently compared to that from the selected deep neural networks, namely the VGG-19 and Inception V3. For each architecture, two models were generated using different wight. The first experiment was conducted using random initialization weight, while the second was pre-trained weight from ImageNet.

As shown in Table 2, the deep neural networks with random initialization generate worse results than traditional machine learning algorithms. The VGG-19 and Inception V3 with random initialization achieve accuracy rates of 61.20% and 61.29%, respectively. The networks seem unable to adjust the weights during training. It is understandable because the number of training images was minimal while the network was deep.

By contrast, the deep neural networks in the form of transfer learning, *i.e.*, the weights are pre-trained from the ImageNet, have shown their superiority in the experiments. Furthermore, the pre-trained Inception V3 produces a better model than the pre-trained VGG-19. It makes sense because Inception V3 has more layers than VGG-19. The number of layers in Inception V3 is 48 layers, while VGG-19 is only 19 layers. By employing a larger number of layers, more complex patterns of images are more likely to be recognized better. Furthermore, the input size of the Inception V3 image is 299 × 299 while that of VGG-19 is 244 × 244. As a result, Inception V3 is able to suppress the loss of important patterns in the images.

Image classification on *Curcuma longa* and *Curcuma zanthorrhiza* can be viewed as a binary classification. Class *Curcuma zanthorrhiza* can be labelled as a positive class because it is the focus of the object being observed. Several performance evaluation metrics in binary classification that are commonly used include precision, recall, and accuracy. Let TP, TN, FP, and FN be True Positive, True Negative, False Positive, and False Negative, respectively. Measurement evaluation of precision, recall, and accuracy can be expressed by;

$$Precision = \frac{TP}{TP + FP} \tag{1}$$

$$Recall = \frac{TP}{TP + FN} \tag{2}$$

$$Accuracy = \frac{TP + TN}{TP + FP + FP + FN} \times 100\%. \tag{3}$$

According to Fig. 6, the precision and recall of the curcuma classification using deep learning are listed as follows in Table 3.

Both models have a similar recall; however, the pre-trained Inception V3 is slightly better than the pre-trained VGG-19. Higher precision means that the model returns more relevant results and fewer irrelevant ones. According to these results, we recommend the Inception V3 with pre-trained weights from the ImageNet to be employed for the curcuma classification.

Our achievement in classifying the curcuma is reasonable for practical use and comparable to other investigations. Similar studies on the product of plant root classification have been reported.

*Elsharif et al. (2020)* investigated potato classification using deep learning and achieved an accuracy of 99.5%. A number of potatoes are classified; however, the colour of the potatoes looks significantly different from each other. Obtaining a high accuracy is possible due to the clear features of the potatoes, although the number of classes used is four (*Elsharif et al., 2020*). *Ramcharan et al. (2017)* investigated cassava disease detection using deep learning. Five diseases were identified and achieved an accuracy of 93.0% (*Ramcharan et al., 2017*). *Mercurio & Hernandez (2019)* utilized CNN to classify sweet potato variety. Five varieties were classified and obtained an accuracy of 96.33% (*Mercurio & Hernandez, 2019*). *Su et al. (2020)* evaluated the quality of potatoes using CNN in six grades. Based on the appearance, the classification had an accuracy of 91.6% (*Su et al., 2020*). *Agustian & Maimunah (2021)* utilized the KNN to classify three rhizomes, namely, temulawak, temuireng, and temumangga. They achieved an accuracy of 87.5% (*Agustian & Maimunah, 2021*). *Puspadhani et al. (2021)* performed onion classification automation using deep learning with the convolutional neural network method. They classified three varieties and achieved an accuracy of 70% (*Puspadhani et al., 2021*). *Shahin et al. (2002)* classified sweet onions based on internal defects using image processing and neural network techniques. They performed binary classification and obtained an accuracy of 80% (*Shahin et al., 2002*). *Khrisne & Suyadnya (2018)* classified herb and spice images into 27 classes and achieved an accuracy rate of 70%. These results are understandable because a number of herbal plants have a fairly high similarity, especially rhizomes. Classification with fewer classes for more specific cases may reduce the error rate. Lastly, *Andayani et al. (2020)* classified turmeric and ginger using based on stomata microscopic images. This work is quite close to what we do, *i.e.*, the class is limited to two plants that have a very high degree of similarity. They classified turmeric and ginger while we classified turmeric and temulawak images. The difference lies in the image used where *Andayani et al. (2020)* used microscopic images while we used mobile phone images. Their experimental results achieved an accuracy of 92.5%. This achievement is quite close to our results. These investigations are summarized in Table 4.

Lastly, a popular issue in computing is running time. Usually, the problem with execution time in deep neural networks is the length of the training time. In curcuma classification, the number of overall images is small, *i.e.*, 647 images. Therefore, the issue of computational time to train the model is not a concern. *Zunair et al. (2021)* showed that the use of transfer learning not only improves classification performance but also requires

**Table 4 Accuracy of classification tasks from other studies.**

| No. | Classification task | #Class | Method | Accuracy | Ref. |
|---|---|---|---|---|---|
| 1 | Potato classification | 4 | CNN | 99.5% | *Elsharif et al. (2020)* |
| 2 | Casava disease detection | 5 | Inception V3 | 93% | *Ramcharan et al. (2017)* |
| 3 | Classification of sweet potato variety | 5 | DCT | 96.33% | *Mercurio & Hernandez (2019)* |
| 4 | Potato quality grading based on depth imaging | 6 | CNN | 91.6% | *Su et al. (2020)* |
| 5 | Rhizome classification | 3 | KNN and SVM | 87.50% | *Agustian & Maimunah (2021)* |
| 6 | Onions classification | 3 | CNN | 70.0% | *Puspadhani et al. (2021)* |
| 7 | Classification of sweet onions | 2 | CNN | 80.0% | *Shahin et al. (2002)* |
| 8 | Herbs and spices recognition | 27 | VGGNet-like | 70% | *Khrisne & Suyadnya (2018)* |
| 9 | Classification of curcuma herbal platns | 2 | CNN | 92.5% | *Andayani et al. (2020)* |
| 10 | Classification of C. Longa & *C. zanthorrhiza* | 2 | Inception V3 | 94.29% | (Ours) |

$6\times$ less computation time and $5\times$ less memory. The runtime should be investigated further with a large number of curcuma images.

## CONCLUSIONS

We carried out image classification of *Curcuma longa* and *Curcuma zanthorrhiza* using two deep neural network architectures, namely, VGG-19 and Inception V3. Applying transfer learning by adopting the weights from ImageNet generates significant improvements for the limited training data. Furthermore, the two deep neural networks with pre-trained weights from ImageNet outperform the traditional models, specifically, the $k$-NN and SVM. The experimental results indicate that Inception V3 generated the best accuracy, *i.e.*, 94.29%. Furthermore, the precision and recall were consecutively 92.4% and 92.8%. The results are still acceptable for practical use. Enhancements are possible to these results, *e.g.*, increasing the images, applying standard treatment for the curcuma, and using more complex transfer learning methods.

## ACKNOWLEDGEMENTS

We thank the PPM team for supporting us in preparing the administrative matters.

### Funding

The article publication cost (APC) was fully funded by Telkom University Research Grant, and there was no additional external funding received for this study. The funders had no role in study design, data collection and analysis, decision to publish, or preparation of the manuscript.

### Grant Disclosures

The following grant information was disclosed by the authors:
Telkom University Research Grant.
## Competing Interests

The authors declare that they have no competing interests.

## Author Contributions

- Agus Pratondo conceived and designed the experiments, performed the experiments, analyzed the data, performed the computation work, prepared figures and/or tables, authored or reviewed drafts of the article, and approved the final draft.
- Elfahmi Elfahmi conceived and designed the experiments, analyzed the data, authored or reviewed drafts of the article, and approved the final draft.
- Astri Novianty conceived and designed the experiments, performed the experiments, prepared figures and/or tables, authored or reviewed drafts of the article, and approved the final draft.

## Data Availability

The data is available at Figshare: Pratondo, Agus (2022): curcuma. figshare. Figure. https://doi.org/10.6084/m9.figshare.19561213.v1.

## Supplemental Information

Supplemental information for this article can be found online at http://dx.doi.org/10.7717/peerj-cs.1168#supplemental-information.

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
