# Peer review of "Classification of Curcuma longa and Curcuma zanthorrhiza using transfer learning"

_PeerJ Computer Science, doi:10.7717/peerj-cs.1168_

## Round 0.1 · original submission · Major Revisions

According to the reviewers' comments, the authors are suggested to revise the paper. The authors need to highlight the modified parts using tracked changes together with the response letter.

Reviewer 2 has suggested that you cite specific references. You are welcome to add it/them if you believe they are relevant. However, you are not required to include these citations, and if you do not include them, this will not influence my decision.

Reviewer 1 ·

Basic reporting

The paper aims to discriminate between images of roots from two related Curcuma, motivated by their culinary/medicinal uses. The paper appears to address a task that can be useful, and a task that I have not seen before. The study goal is well-defined and well-constrained.

An important problem throughout is that the authors do not make clear the difference between "transfer learning", which is using the network state trained from one task to assist with training on another task, and the reuse of an existing neural network architecture (whether pretrained or not).
For example in the abstract (line 17): "transfer learning, specifically VGG-19 and Inception V3". VGG-19 and Inception V3 are not transfer learning - they are neural network architectures. This same confusion appears around line 119 onwards. The authors must clarify their ideas here: VGG-19 and Inception V3 are not transfer learning algorithms. Transfer learning only occurs when those networks have been pre-trained on some other dataset. The authors MUST specify which datasets they are pretrained on; without this, the reporting is seriously lacking. (The pretraining has usually been done by someone else before the NN is uploaded.)

Related literature on turmeric images appears to be surveyed well, as well as the basic description of the plant species considered; this is not my domain. I would like the "proven" claims in the paragraph starting on line 46 to be cited directly to the relevant scientific literature. I see 2 citations at the very end of the paragraph, but it is not clear which claims are supported by which papers. More direct citing of the claims would be better.

Table 4 (accuracy from other studies) is impossible to interpret, unless we know how many classes were considered in each of those studies. Authors must add a column to indicate this. They also need to fill in the number for their own study.

Precision and recall can be quoted as percentages too, i.e. change "0.921" to "92.1%".

The acknowledgment section needs changing. "So long and thanks for all the fish."

I was able to find 2 papers that seem highly relevant. The authors should include them in their literature review, or justify why not:

(a) Khrisne et al (2018) "Indonesian Herbs and Spices Recognition using Smaller VGGNet-like Network" 10.1109/ICSGTEIS.2018.8709135 -- this uses photos of multiple plants including turmeric, and also a very similar NN architecture to the present VGG-19.

(b) Andayani et al (2020) "The Implementation of Deep Learning Using Convolutional Neural Network to Classify Based on Stomata Microscopic Image of Curcuma Herbal Plants" 10.1088/1757-899X/851/1/012035 --- this uses exactly the two species studied in the present paper. It is different because it uses microscopy images rather than smartphone photos, but it is a similar method and a possible alternative strategy to the one in the present paper, so it should be discussed.

Experimental design

The machine learning aspects of the study appear broadly to be correct. The two-class classification scenario (with no other "distractor" classes) is a definite limitation, but I assume that the use case here is that the method would only be used with images of suspected turmeric, so the limitation is not unrealistic.

The confusion between "transfer learning" and "using an existing architecture" also affects the experimental setup. The authors compare kNN and SVM as baseline models, against the (presumably pretrained) NNs. Thus, the problem is that there are two different factors being modified at the same time: (a) the choice of ML algorithm, (b) the use of transfer learning (pretraining) from some other dataset(s).
The authors should fix this by running a test of the two NNs (VGG-19 and Inception V3) WITHOUT pretraining, i.e. with randomly-intiialised NN parameters. This will

For SVM and kNN, what are the features input to the algorithms? Are the raw high-dimensional pixel values used? Is this appropriate? It seems unlikely that kNN and SVM would be used with raw pixel data -- previous work would presumably extract some features first. Authors should clarify. They should ideally also check whether their kNN/SVM method is really a good baseline, i.e. if the feature representation is appropriate as seen in previous literature.

Validity of the findings

The validity of the findings depends somewhat on whether we consider the goal to be "reliable discrimination of the two image classes" or "reliable discrimination of the two image classes using transfer learning" (as claimed in the title/abstract). The discrimination task appears to be successfully achieved, but the authors have not demonstrated which factor is creating the good accuracy.

Reviewer 2 ·

Basic reporting

In this manuscript, the authors use pre-trained deep learning models to perform image classification of Curcuma longa and Curcuma zanthorrhiza. With the exception of some grammatical errors, the paper is relatively well written and technically sound, though the introduction section provides a deficient analysis of related approaches in the literature. Among the missing references:

[1] STAR: Noisy Semi-Supervised Transfer Learning for Visual Classification; Proc. International Workshop on Multimedia Content Analysis in Sports, 2021
[2] Melanoma detection using adversarial training and deep transfer learning; Physics in Medicine & Biology, 2020.

Experimental design

The experimental results demonstrate that pre-trained models outperform traditional machine learning algorithms.

Validity of the findings

Since pre-trained models depend on several parameters that need to be fine-tuned, it is, however, unclear how the choice of such parameters would affect the overall performance of these models in comparison with the traditional baselines. Also, the runtime analysis needs to be included either in the results or discussions section.

---

## Round 0.2 · Minor Revisions

Please revise the paper again according to the reviewers' comments.

Reviewer 1 ·

Basic reporting

The paper is generally complete and acceptable, and improved from the first submission. The scientific writing standard is a little below standard in some places.

Improvements to make:

In the "Results" section, there should NOT be any new method introduced. Hence, the whole "Experiment settings" subsection (lines 147 to 169) should be moved from its current position up to the end of the "Method" section, i.e. after the current line 142. Once that is done, there is no need for the introductory sentences lines 144-146 so they can be deleted.

Similarly, lines 191 to 193 in the discussion tell us how the RGB pixel values were used in the baseline systems. This is also a method detail and so should be in the "Method" section.

Missing information: The authors describe their use of data augmentation, but they fail to mention how large is the "expanded" dataset created by augmentation. This should be stated. For example, it is common in experiments to use data augmentation to create 10 times, or 100 times, as many items as in the original dataset.


Minor/typos:
l16: "aims to build a model" -- be clearer, change "model" to "classifier"
l19: "ImangeNet"
l76: "two classes, namely, (Curcuma longa) and temulawak (Curcuma zanthorrhiza)" -- a word seems to be missing after "namely".
l93: "The first one is chosen since it is quite old. However, the algorithm is still commonly used" -- it is bizarre to suggest that the algorithm is chosen because it is quite old. Simply shorten this text to "The first one is chosen since it is commonly used"
l206 "Contrastly" -> "By contrast"
l223 "than irrelevant" -> "and fewer irrelevant"

Experimental design

It is very good that the authors added the random-initialisation tests to their paper. From this we can see that the transfer learning is indeed important to enable deep learning to be used for their chosen task given the current data availability.

Validity of the findings

The baseline results, using the traditional classifiers, have low external validity, since the authors choose to use raw pixel features whereas most existing projects would use feature extraction before kNN or SVM.

The authors discuss this (line 191 onwards), claiming that it would be unfair to engineer features for one algorithm and not another. I do not agree with this: deep learning is widely described as incorporating feature extraction into its early layers, and thus acting as a replacement for BOTH the feature extraction and the classifier.

My concern here relates to the validity of the baseline, and not to the validity of the DL algorithms used as the main topic. The main method proposed is indeed evaluated in a valid manner for the chosen use case.

Additional comments

The paper is now more complete, valid and readable than the first submission. The authors have made honest effort to account for the feedback from the reviews, and the paper is better as a result.

Reviewer 2 ·

Basic reporting

The revised paper is relatively well written and technically sound.

Experimental design

The experimental results demonstrate the effectiveness of the proposed approach.

Validity of the findings

Comparisons with existing methods demonstrate the superiority of the proposed method.

---

## Round 0.3 · accepted · Accept

The revision seems good. No more comments.

Reviewer 1 ·

Basic reporting

Thanks for making the changes. Looks good now.

Experimental design

All acceptable (depsite my concerns with the baseline, as expressed last time)

Validity of the findings

All acceptable